

# Six-minute rowing test: a practical tool for training prescription, from ventilatory thresholds and power outputs, in amateur male rowers

Álvaro Huerta Ojeda[1] and  Miguel Riquelme Guerra[2,3]

[1] Grupo de Investigación en Salud, Actividad Física y Deporte ISAFYD, Universidad de Las Américas, Viña del Mar, Chile
[2] Magíster en Medicina y Ciencias del Deporte, Facultad de Ciencias, Universidad Mayor, Santiago, Chile
[3] Escuela de Kinesiología, Facultad de Salud, Universidad Santo Tomás, Viña del Mar, Chile

Corresponding author
Álvaro Huerta Ojeda,
achuertao@yahoo.es

## ABSTRACT

**Background**. The 6-minute rowing ergometer test ($6\text{-min}_{RT}$) is valid and reliable for establishing maximal aerobic power (MAP) in amateur male rowers. However, ventilatory thresholds (VTs) have not yet been established with their mechanical correspondence in this test.

**Objective**. The primary objective was to determine the VTs in the $6\text{-min}_{RT}$ achieved by amateur male rowers, while the secondary objective was to determine the correspondence between ventilatory, mechanical, and heart rate (HR) outcomes of the $6\text{-min}_{RT}$.

**Methods**. Sixteen amateur male rowers were part of the study. All participants were instructed to perform an incremental test (IT) and a $6\text{-min}_{RT}$. Determination of the ventilatory parameters for the first ventilatory threshold (VT1), the second ventilatory threshold (VT2), and $6\text{min}_{RT}VO_{2max}$ were performed by correlating the outcomes of VT1, VT2, and $VO_{2max}$ obtained in the IT, with the outcomes of $6\text{-min}_{RT}$. For these purposes, Pearson's test was used, with the following criteria: trivial, <0.1; small, 0.1–0.3; moderate, 0.3–0.5; high, 0.5–0.7; very high, 0.7–0.9; or practically perfect, >0.9. The significance level was $p < 0.05$.

**Results**. The IT analysis determined that VT1 and VT2 correspond to 55 and 80% of $VO_{2max}$, respectively. A high correlation was observed between IT outcomes in VT1, VT2, and $VO_{2max}$, with the outcomes of $6\text{-min}_{RT}$ ($r > 0.6$).

**Conclusion**. Based on IT ventilatory parameters and concordance analysis, VT1 and VT2 of $6\text{-min}_{RT}$ are determined at 55 and 80%, respectively, of both ventilatory parameters and their corresponding mechanical outcomes and HR.

## INTRODUCTION

Some factors that determine the performance of the rowers are physical and technical development, good boat stabilization, and correct synchronization between the members of the boat (*Muniesa & Díaz, 2010*; *Huntsman, Drury & Miller, 2011*; *Rich, Pottratz & Leaf, 2020*). Considering that all regattas are outdoors, factors to consider for the good

performance of the rowers are the environment, the environmental temperature, the wind, and the swell also condition the performance in this sport (*Ingham et al., 2002*). From an energetic point of view, scientific evidence has shown that the contribution of aerobic metabolism fluctuates between 70–88%, while anaerobic metabolism fluctuates between 12–30% (*Hagerman et al., 1978*; *Secher, 1993*; *Pripstein et al., 1999*). Because the aerobic component is a determining factor in the performance of rowers, most of the research has focused on this component (*Klusiewicz et al., 2016*; *Das et al., 2019*). It is also important to mention that most of the direct evaluations of maximum oxygen consumption ($VO_{2max}$) in rowers, which are considered the gold standard, have been developed in the elite category (*Turnes et al., 2020*), mainly due to the high cost of gas analysis in laboratories (*Wagner, 1996*). The high cost of these evaluations, which affects different sports (*Metaxas et al., 2005*), has conditioned coaches to prescribe training loads based on the results of field tests (indirect). In this context, there is evidence that field tests have low reliability in prescribing exercise to amateur rowers (*Huerta Ojeda et al., 2022a*).

To determine the ventilatory parameters in rowers (aerobic component), specifically the $VO_{2max}$, the gold standard used is the incremental test (IT) on the rowing machine (*Mekhdieva, Zakharova & Timokhina, 2019*). In parallel, during any test (especially during the IT), the rowing meter allows for measuring mechanical parameters, including power output (PO) (*Bourdin, Messonnier & Lacour, 2004*), a physical determinant factor in the performance of rowers (*Bourdin et al., 2017*). PO on the rowing ergometer results from the stroke rate and the force exerted by the rower in each stroke (*Huerta Ojeda et al., 2022a*). When $VO_{2max}$ is reached during an incremental treadmill test, an ergometric equivalent in kilometers per hour, defined as maximal aerobic speed (MAS), is also reached (*Howley, Bassett & Welch, 1995*). In the case of amateur rowers, there is evidence that, in addition to reaching $VO_{2max}$ during an incremental test on a rowing machine, they also reach an equivalent PO, defined as maximal aerobic power (MAP) (*Bourdin et al., 2017*; *Huerta Ojeda et al., 2022b*). This MAP is a critical intensity for the prescription of training loads and tactical preparation for regattas (*Bourdin et al., 2017*). Another determining factor in observing the physical level, programming the training loads, and monitoring the progress of the rowers is the ventilatory thresholds (VTs) (*Davis, 1985*; *Cerezuela-Espejo et al., 2018*). In this context, the scientific literature describes a first ventilatory threshold (VT1), which is identified with the first increase in the ventilatory equivalent of oxygen (pulmonary ventilation/oxygen consumption ($VE/VO_2$)) without a simultaneous increase in the ventilatory equivalent of carbon dioxide ($VE/VCO_2$). In contrast, the second ventilatory threshold (VT2) is identified with an increase in $VE/VO_2$ and $VE/VCO_2$ and a decrease in carbon dioxide ($CO_2$) at end-expiration ($PETCO_2$) (*Cerezuela-Espejo et al., 2018*). In fact, VTs are indicators of the ability of athletes to exercise at specific intensities for prolonged periods (*Billat, 1996*). Moreover, VTs are attributed to the specificity of training loads since these, specific to each athlete, increase the oxidative capacity of muscle fibers and significantly improve the cardiorespiratory system (*Mickelson & Hagerman, 1982*). Therefore, when $VO_{2max}$ is evaluated through an IT on the rowing machine, along with obtaining the VTs, their mechanical equivalences are also observed (*Cerezuela-Espejo et al., 2018*; *Huerta Ojeda et al., 2022a*). In this sense, there have been several

attempts to validate non-invasive and practical methods to help determine VTs (*Cabo, Martinez-Camblor & del Valle, 2011*; *Stefanov & Neykov, 2021*). This background shows the importance of establishing different training zones based on ventilatory parameters (*López Chicharro, Vicente Campos & Cancino López, 2013*). However, most non-invasive methods for assessing VTs in amateur rowers do not simultaneously integrate ventilatory and mechanical parameters, resulting in low reliability of exercise prescription in this category (*Huerta Ojeda et al., 2022a*).

Intending to contribute with a scientific background in amateur rowers, we described and analyzed the kinetics of ventilatory and mechanical parameters in male amateur rowers in a recent study. Male amateur rowers reached $VO_{2max}$ and MAP between 330–345 seconds (s) (*Huerta Ojeda et al., 2022a*). Subsequently, based on the study described above, we determined the validity and reliability of a 6-$min_{RT}$ as a predictor of PO in male amateur rowers. At the end of the research and after comparing the kinetics of ventilatory and mechanical parameters between a TI and the 6-minRT, we concluded that the 6-min $_{RT}$ is a valid and reliable test for establishing MAP in male amateur rowers (*Huerta Ojeda et al., 2022b*). Despite the documented evidence, it has not yet been determined at what percentage of MAP the VTs are found when the 6-min $_{RT}$ isapplied, with their corresponding mechanical equivalence. Consequently, this study aimed to determine amateur rowers' power and cardio-respiratory characteristics at each VT during a 6-$min_{RT}$. The secondary objective was to determine the correspondence between ventilatory, mechanical, and HR outcomes of the 6-$min_{RT}$.

## MATERIALS & METHODS

### Experimental approach to the problem

An observational design with an associative and cross-sectional strategy was used to relate the outcomes of VT1, VT2, and $VO_{2max}$ obtained in the IT to the outcomes of the 6-$min_{RT}$ (*Ato, López & Benavente, 2013*) (Fig. 1). All study participants attended the laboratory for three days at 72-hour intervals. Also, participants did not exercise between assessment days. The time between races ensures the rowers' physical recovery (*American College of Sports Medicine, 2013*). During the first visit, participants signed the informed consent form and were evaluated with basic anthropometry; on the second day, they performed the IT, and on the third day, the 6-$min_{RT}$ (Fig. 1A).

### Participants

Sixteen male amateur rowers participated voluntarily in this study (age $20.8 \pm 2.2$ years [range: 18–27], body mass $79.7 \pm 7.6$ kg [range: 68.2–103.6], stature $176.2 \pm 5.1$ cm [range: 169–186], BMI $25.7 \pm 2.2$ kg/m$^2$ [range: 22.5–31.3], body fat percentage $12.9 \pm 2.9\%$ [range: 7.1–19.9], $VO_{2max}$ $46.32 \pm 4.09$ mLO$_2$ kg$^{-1}$ min$^{-1}$ [range: 36.48–52.66], and theoretical maximum HR (*Tanaka, Monahan & Seals, 2001*) $191.5 \pm 1.9$ [range: 187.1–193.4]. The inclusion criteria were previously described in *Huerta Ojeda et al. (2022b)*. Specifically, the main inclusion criterion was the $VO_{2max}$ assessed in the IT. This initial evaluation made it possible to verify the fitness level of the male amateur rowers. In this regard, all participants in the study presented a $VO_{2max}$ between 33–55 mLO$_2$ kg$^{-1}$ min$^{-1}$. Also, it was found

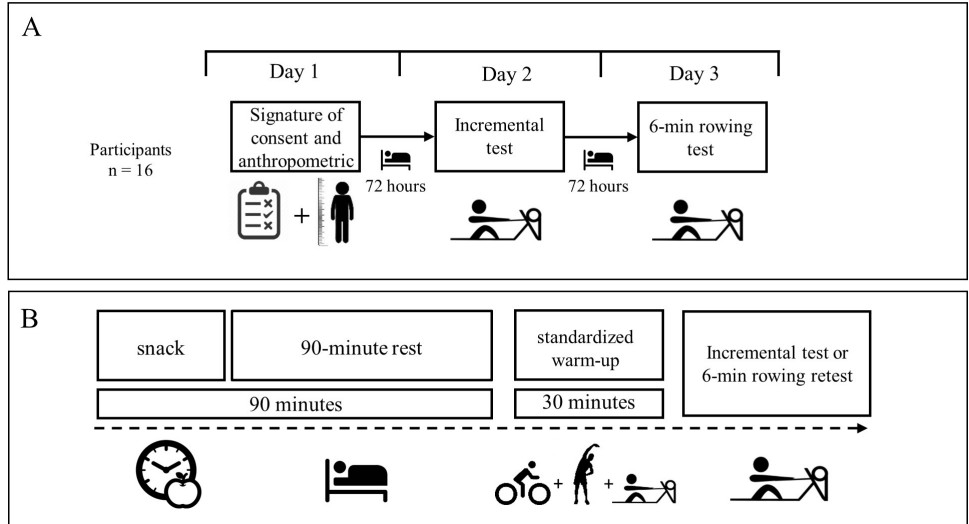

**Figure 1  Research design.**

that the rowers trained between 3–6 h per week. All these antecedents allowed defining the participants as amateur rowers, also classified as 'Recreationally Active' (*McKay et al., 2022*). In contrast, rowers who did not reach the $VO_{2max}$ in the IT, according to the criteria described in the literature (*Howley, Bassett & Welch, 1995*; *American College of Sports Medicine, 2013*), or could not finish the 6 min of the 6-min$_{RT}$ were eliminated during the study. Statistical software (G*Power, v3.1.9.7; Heinrich-Heine-Universität, Germany) was used to calculate the sample (*Faul, 2020*). The combination of tests used in the statistical software to calculate the sample size was as follows: (a) $t$-test, (b) Correlation: Point biserial model, and (c) A priori: Compute required size –given $\alpha$, power, and effect size. Tests considered two tails, slope H1 = 0.61, $\alpha$-error < 0.05, and a desired power (1 − $\beta$ error) = 0.8. The total sample size was 16 participants. All participants were informed of the study's objective and the possible experiment risks. Before applying the protocols, all amateur rowers signed the informed consent form in person. The study and the informed consent were approved by the Scientific Ethical Committee of the Universidad Mayor, Santiago, Chile (registration number: 197_2020) and developed under the ethical standards for exercise and sports sciences (*Harriss, Macsween & Atkinson, 2019*).

## Anthropometric measurements

For the characterization of the sample, body mass, stature, body mass index (BMI), and body fat percentage were evaluated. According to the international anthropometric assessment protocol, weight (kg) was assessed with a digital weight (TANITA, model InnerScan BC-554®, Tokyo, Japan). Height (m) was assessed with a stadiometer (SECA, model 700®; SECA, Hamburg, Germany), respecting the Frankfort plane and maximum inspiration. This ensured that the participants were evaluated in a euhydrated condition. The body fat percentage was assessed using an impedance meter (Tanita Inner Scan, BC-554® digital scale; Tanita, Tokyo, Japan).

## Snack

The snack prevents athletes from starting evaluations with a low blood glucose level (*Ojeda et al., 2019*). Este snack ensured a blood glucose level between 100–120 mg/dl, considered optimal before exercise (*DeMarco et al., 1999*). It was a carbohydrate load before the IT and 6-min$_{RT}$. All participants were available two hours before the tests in a fasting condition. The snack consisted of 2 g of rapidly absorbed carbohydrates per kg body mass, including fruit, fruit juice, bread, and jam (Fig. 1B).

## Standardized warm-up

The warm-up consisted of 10 min on a bicycle (Airbike Xebex® resistance; ABMG-3 Xebex Fitness). The warm-up intensity was 60–70% of the theoretical maximum HR, calculated through the formula: 208-(0.7*age) (*Tanaka, Monahan & Seals, 2001*). Five minutes of upper and lower extremity ballistic movements were then included (upper extremity: $1 \times 10$ shoulder adductions, abductions, flexion, and extension; lower extremity: $1 \times 10$ hip adductions, abductions, flexion and extension, and knee and $1 \times 10$ ankle flexion and extension, right and left respectively). After, athletes rowed for five minutes between 60–70% of the theoretical maximum HR (*Tanaka, Monahan & Seals, 2001*), and finally, there was a 10-minute rest (this time was used to install the mask and HR sensors) (Fig. 1B).

## Incremental test

The objective and characteristics of the test were previously described in *Huerta Ojeda et al. (2022b)*. Specifically, this test aims to progressively reach VO$_{2max}$ and its equivalent PO or MAP. All ventilatory parameters were evaluated during IT with an automatic gas analyzer system (model Quark CPET; Cosmed, Rome, Italy). Before IT, the analyzer was calibrated strictly according to the manufacturer's recommendations. The data were processed through a laptop computer, which calculated the results using software developed by the manufacturer. The IT was performed on a remoergometer (Concept2 Model D, PM5 monitor) using a drag factor of 117–119. The IT started with 100 watts (W) and was increased by 35 W every minute until exhaustion or the impossibility of maintaining the requested power (*Mekhdieva, Zakharova & Timokhina, 2019*). Two experienced investigators monitored the test. VO$_{2max}$ was determined based on the following criteria (*Howley, Bassett & Welch, 1995*; *American College of Sports Medicine, 2013*): (i) when setting plateau, display a variation $< 150$ mlO$_2$ min$^{-1}$, (ii) a respiratory exchange ratio (RER) $\geq 1.1$, and/or (iii) Theoretical maximum HR $\geq 90\%$. VO$_{2max}$ was recorded with absolute (LO$_2$ min$^{-1}$) and relative values (mlO$_2$ kg$^{-1}$ min$^{-1}$). In addition, pulmonary ventilation was recorded (VE) at VO$_{2max}$ (L min$^{-1}$) and RER at VO$_{2max}$ (VCO$_2$/VO$_2$) (*Howley, Bassett & Welch, 1995*). VT1 and VT2 were identified according to the following criteria (*Davis, 1985*; *Lucía et al., 2000*; *Cerezuela-Espejo et al., 2018*): VT1, the intensity causing the first increase in ventilatory oxygen equivalent (VE/VO$_2$) without a simultaneous increase in ventilatory carbon dioxide equivalent (VE/VCO$_2$); VT2, the intensity that causes an increase in VE/VO$_2$ and VE/VCO$_2$ and a decrease of CO2 at the end of exhalation (PETCO$_2$). All the mechanical parameters performed in the rowing meter were recorded during IT. Also, the HR was recorded during the development of the IT

through an HR monitor (model H10[®], Polar, Kempele, Finland). During IT, ventilatory parameters were recorded breath by breath, while mechanical parameters and HR were recorded from stroke to stroke. Subsequently, all data were averaged and synchronized at 15 s intervals. At the end of the IT, to evaluate the rating of perceived exertion (RPE) of the participants, the modified Borg scale was used (1–10) (*Borg, 1990*).

### Six-minute rowing test

The purpose of this test is that the participant rows the longest distance possible for 6 min on an ergometer rowing machine. From a ventilatory and mechanical point of view, the 6-min$_{RT}$ is a maximal test that allows the establishment of the MAP in amateur rowers (*Huerta Ojeda et al., 2022b*). The final 6-min$_{RT}$ distance is recorded in meters (m). The 6-min$_{RT}$ was performed on a rowing machine (Concept2 Model D, monitor) using a drag factor of 117–119. Two experienced investigators monitored the test. During the 6-min$_{RT}$, all ventilatory parameters were evaluated with the same automatic gas analyzer system used in the IT (model Quark CPET; Cosmed, Rome, Italy). During the 6-min$_{RT}$, ventilatory parameters were recorded breath-by-breath and then averaged at 15-s intervals. In this study, the slow and fast $VO_2$ components were determined by visual inspection of each curve, specifically from the moment a plateau was generated in the $VO_2$ curve until the end of the 6-min$_{RT}$ (*Billat et al., 1998*; *Womarck et al., 2000*). For the statistical analysis, the ventilatory parameters correspond to the slow component of the $VO_2$ ($VO_{2max}$ abs, $VO_{2max}$ rel, VE, and RER). During the 6-min$_{RT}$, all mechanical parameters performed on the rowing meter were recorded. Also, HR was recorded during the 6-min$_{RT}$ through an HR monitor (model H10[®]; Polar). During the 6-min$_{RT}$, mechanical parameters and HR were recorded from stroke to stroke. Subsequently, all data were averaged and synchronized in a single 360 s interval. At the end of the 6-min$_{RT}$, to evaluate the rating of perceived exertion (RPE) of the participants, we used the modified Borg scale (1–10) (*Borg, 1990*).

### Statistical analyses

Data were collected as described in *Huerta Ojeda et al. (2022b)*. Specifically, ventilatory, mechanical, and HR parameters were sorted for all the tests on a spreadsheet designed for the study. Descriptive data are presented as means and standard deviation (SD). The Shapiro–Wilk test confirmed the normal distribution of the data ($p > 0.05$). Pearson's test calculated the correlation between the VT1, VT2, and $VO_{2max}$ outcomes obtained in the IT with the 6-min$_{RT}$ outcomes ($r$ and $r^2$) (*Hopkins et al., 2009*). The criteria for interpreting the strength of the r coefficients were as follows: trivial (<0.1), small (0.1–0.3), moderate (0.3–0.5), high (0.5–0.7), very high (0.7–0.9), or practically perfect (>0.9). A 95% confidence interval was used for all statistical analyses, and the significance level was $p < 0.05$. All statistical analyses were performed with Prism version 7.00 for Windows[®] software.

## RESULTS

The first analysis shows that during IT, the amateur rowers in the present study reached a $VO_{2max}$ equivalent to $3.67 \pm 0.27$ L$O_2$ min$^{-1}$. In parallel, in the 6-min$_{RT}$, a $VO_{2max}$ was

**Table 1  Ventilatory thresholds and maximal oxygen uptake in Incremental Test and 6-minute rowing test ($n = 16$).**

| Variables | VT1 (mean ± SD) | VT2 (mean ± SD) | $VO_{2max}$(mean ± SD) | 6-min$_{RT}$ |
|---|---|---|---|---|
| $VO_2$ absolute ($LO_2 \cdot min^{-1}$) | $2.03 \pm 0.15$ | $2.94 \pm 0.21$ | $3.67 \pm 0.27$ | $3.68 \pm 0.19$ |
| $VO_2$ absolute (% of $VO_{2max}$) | 55% | 80% | 100% | — |
| $VO_2$ relative ($LO_2 \cdot min^{-1}$) | $25.49 \pm 2.84$ | $37.17 \pm 3.60$ | $46.32 \pm 4.09$ | $46.55 \pm 4.67$ |
| $VO_2$ relative (% of $VO_{2max}$) | 54% | 80% | 100% | — |
| VE ($L \cdot min^{-1}$) | $48.9 \pm 9.8$ | $84.0 \pm 8.3$ | $150.1 \pm 11.8$ | $137.7 \pm 9.9$ |
| VE (% of $VO_{2max}$) | 32% | 56% | 100% | — |
| RER ($VCO_2/VO_2$) | $0.87 \pm 0.08$ | $0.98 \pm 0.06$ | $1.25 \pm 0.05$ | $1.14 \pm 0.04$ |
| RER (% of $VO_{2max}$) | 62% | 78% | 100% | — |
| Distance (m) | — | — | — | $1,697.3 \pm 39.9$ |
| Power output (W) | $121.2 \pm 14.5$ | $212.7 \pm 26.6$ | $391.9 \pm 29.9^{*}$ | $292.0 \pm 19.7$ |
| Power output (% of MAP) | 37.9% | 66.5% | 100% | — |
| Pace every 500 m (min:s) | $2:22.4 \pm 6.1$ | $1:58.1 \pm 4.9$ | $1:40.3 \pm 3.4$ | $1:46.7 \pm 2.5$ |
| Pace (% of $VO_{2max}$) | 62% | 85% | 100% | — |
| Stroke rate (spm) | $20.2 \pm 3.9$ | $23.6 \pm 4.0$ | $31.1 \pm 3.3$ | $30.4 \pm 1.9$ |
| HR (bpm) | $132.9 \pm 5.5$ | $162.7 \pm 10.6$ | $191.2 \pm 6.8$ | $193.8 \pm 7.2$ |
| HR (% of $HR_{max}$) | 69%[**] | 85%[**] | 99%[**] | — |
| RPE (Borg: 1-10) | — | — | $7.94 \pm 0.57$ | — |

Notes.

bpm, beat per minute; HR, heart rate; $HR_{max}$, maximum teorical heart rate; $L \cdot min^{-1}$, liters per minute; $LO_2 \cdot min^{-1}$, liters of oxygen per minute; $mLO_2 \cdot min^{-1}$, milliliters of oxygen per kilogram per minute; m, meters; MAP, maximal aerobic power; min, minutes; PO, power output; RER, respiratory exchange ratio; RPE, rating of perceived exertion; s, seconds; SD, standard deviation; spm, strokes per minute; VE, minute ventilation; $VO_2$, oxygen uptake; $VCO_2/VO_2$, carbon dioxide production/maximal oxygen uptake; $VO_{2max}$, maximal oxygen uptake; W, watts; VT1, aerobic threshold; VT2, anaerobic threshold; 6-min$_{RT}$, 6-minute rowing test.

[*]corresponding to MAP.

[**]corresponding to % of $HR_{max}$.

equivalent to $3.68 \pm 0.19$ $LO_2$ $min^{-1}$. The ventilatory, mechanical, and HR parameters of IT and 6-min$_{RT}$ are listed in Table 1.

The analyses of concordance between $VO_2$ and PO for ventilatory thresholds and $VO_{2max}$, evaluated in the IT, were as follows: VT1, $r = 0.40$ (95% CI [$-0.11$–$0.74$], $p = 0.12$); VT2, $r = 0.52$ (95% CI [$0.03$–$0.80$], $p = 0.038$); and $VO_{2max}$, $r = 0.56$ (95% CI [$0.09$–$0.82$], $p = 0.023$). The correlations of these parameters are reported in Fig. 2.

The agreement values between $VO_2$, distance, and PO in the 6-min$_{RT}$, evaluated in the 6-min$_{RT}$, were as follows: $VO_2$ rel *vs.* distance, $r = 0.69$ (95% CI [$0.31$–$0.88$], $p = 0.0026$); $VO_2$ rel *vs.* PO, $r = 0.70$ (95% CI [$0.31$–$0.88$], $p = 0.0025$); and PO *vs.* distance, $r = 0.99$ (95% CI [$0.99$–$0.99$], $p < 0.0001$). The correlations of these parameters are reported in Fig. 3.

The concordance values between VT1, VT2, and $VO_{2max}$ of the IT and the outcomes of the 6-min$_{RT}$ are reported in Table 2.

## DISCUSSION

This study was designed to determine the VTs in the 6-min$_{RT}$achieved by amateur male rowers and the correspondence between the ventilatory, mechanical, and HR outcomes of the 6-min$_{RT}$. The correlation analysis showed that there is a high correlation ($r < 0.6$) between the main ventilatory parameters ($VO_2$), mechanical (PO), and HR obtained in

Peer J

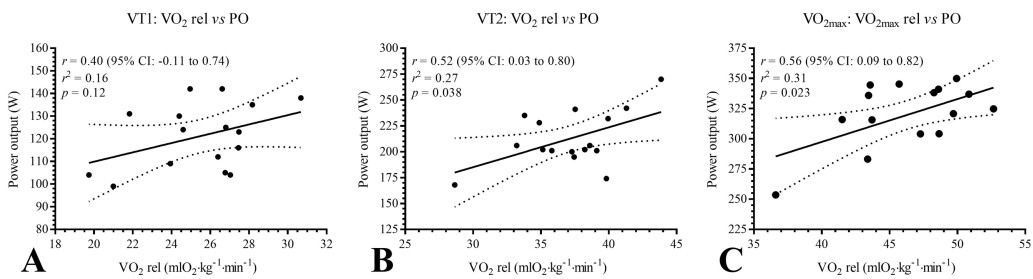

**Figure 2** **(A–C) Correlation coefficient between oxygen consumption and power output in VT1, VT2 and VO$_{2max}$ evaluated in the Incremental Test.** mlO$_2$ kg$^{-1}$ min$^{-1}$, milliliters of oxygen per kilogram per minute; *p*, *p*-value; PO, power output; VO$_{2max}$, maximal oxygen uptake; VO$_2$ rel, oxygen uptake relative; *r*, correlation coefficient; r$^2$, squared correlation coefficient; VT1, aerobic threshold; VT2, anaerobic threshold; W, watts; 95% CI, 95% Confidence Intervals.

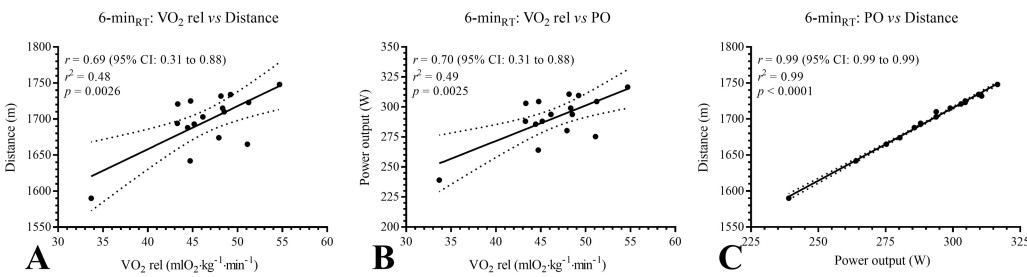

**Figure 3** **(A–C) Correlation coefficient between oxygen consumption, distance and power output in 6-min$_{RT}$.** m, meters; mlO$_2$ kg$^{-1}$ min$^{-1}$, milliliters of oxygen per kilogram per minute; *p*, *p*-value; PO, power output; VO$_2$ rel, oxygen uptake relative; *r*, correlation coefficient; r$^2$, squared correlation coefficient; W, watts; 6-min$_{RT}$, 6-minute rowing test; 95% CI, 95% Confidence Intervals.

the IT, both in VTs and VO$_{2max}$, and some of the outcomes of the 6-min$_{RT}$ (distance, VO$_{2max}$, PO, and HR). These results suggest the following: (a) the 6-min$_{RT}$ allows to reach the VO$_{2max}$ and that the PO of this test can be considered as MAP (*Huerta Ojeda et al., 2022b*), and (b) that the results of r$^2$ between ventilatory, mechanical and HR parameters of both tests (IT and 6-min$_{RT}$) determine that VT1 and VT2 are at 55 and 80% of 6-min$_{RT}$.

## VTs and VO$_{2max}$ through IT

When analyzing VT1, it was observed that male amateur rowers reached a VO$_2$ equivalent to 55% of VO$_{2max}$. From the ventilatory perspective, the percentage value of VO$_{2max}$ reached by the present study's participants is lower than existing theoretical reference parameters (60–65% of VO$_{2max}$) (*López Chicharro, Vicente Campos & Cancino López, 2013*). Finally, the low percentage of VT1 observed may be associated with the low aerobic capacity of the amateur rowers in the study (*Billat, 1996*). This situation would limit both the oxidative capacity of muscle fibers and the aerobic contribution to energy metabolism (*Mickelson & Hagerman, 1982*). However, this point needs to be studied further. Another essential variable to analyze to establish VT1 is the RER (*Davis, 1985*; *Lucía et al., 2000*; *Cerezuela-Espejo et al., 2018*). In this sense, the value of RER observed in the present

**Table 2 Correlation coefficient between ventilatory thresholds and 6-min$_{RT}$ outcomes ($n = 16$).**

| | | 6-min$_{RT}$ | | | | | | |
|---|---|---|---|---|---|---|---|---|
| | | Distance | PO | VO$_2$ rel | VO$_2$ abs | VE | RER | HR |
| VT1 | PO | 0.49 | 0.47 | 0.38 | 0.36 | −0.28 | −0.17 | −0.23 |
| | VO$_2$ rel | 0.64[†] | 0.64[†] | 0.78[‡] | 0.49[+] | 0.29 | 0.27 | 0.10 |
| | VO$_2$ abs | 0.42 | 0.42 | 0.17 | 0.59[+] | 0.05 | −0.01 | 0.04 |
| | VE | 0.20 | 0.17 | 0.12 | 0.27 | −0.26 | −0.30 | −0.30 |
| | RER | −0.14 | −0.15 | −0.24 | −0.00 | −0.08 | 0.02 | −0.02 |
| | HR | −0.62[†] | −0.63[†] | −0.64[†] | −0.33 | −0.03 | 0.07 | −0.03 |
| VT2 | PO | 0.66[†] | 0.65[†] | 0.55[+] | 0.53[+] | −0.04 | −0.06 | −0.45 |
| | VO$_2$ rel | 0.67[†] | 0.66[†] | 0.82[f] | 0.53[+] | 0.29 | 0.30 | 0.11 |
| | VO$_2$ abs | 0.37 | 0.36 | 0.11 | 0.60[+] | −0.01 | −0.07 | 0.08 |
| | VE | 0.18 | 0.19 | 0.09 | 0.45 | 0.31 | −0.11 | 0.00 |
| | RER | −0.30 | −0.30 | −0.43 | −0.12 | 0.05 | −0.04 | 0.14 |
| | HR | −0.18 | −0.18 | −0.32 | −0.29 | 0.06 | −0.15 | 0.25 |
| VO$_{2max}$ | PO | 0.85[f] | 0.84[f] | 0.57[+] | 0.29 | 0.25 | 0.13 | −0.04 |
| | VO$_2$ rel | 0.68[†] | 0.68[†] | 0.76[f] | 0.49 | 0.32 | 0.29 | 0.01 |
| | VO$_2$ abs | 0.29 | 0.30 | −0.0 | 0.50[+] | −0.01 | −0.10 | −0.02 |
| | VE | 0.37 | 0.37 | 0.41 | 0.13 | 0.77[‡] | 0.37 | 0.30 |
| | RER | −0.04 | −0.05 | 0.03 | −0.38 | 0.41 | 0.55[+] | 0.50[+] |
| | HR | −0.37 | −0.36 | −0.37 | −0.61[+] | 0.40 | 0.43 | 0.78[‡] |

Notes.

HR, heart rate; PO, power output; RER, respiratory exchange ratio; VE, minute ventilation; VO$_{2max}$, maximal oxygen uptake; VO$_2$ rel, oxygen uptake relative; VO$_2$ abs, oxygen uptake absolute; VT1, aerobic threshold; VT2, anaerobic threshold.

[+] $p < 0.05$.
[†] $p < 0.01$.
[‡] $p < 0.001$.
[f] $p < 0.0001$.

study for VT1 ($0.87 \pm 0.08$ VCO$_2$/VO$_2$) is coincident with the RER value reported by other research in rowers ($0.89 \pm 0.018$ VCO$_2$/VO$_2$) (*Cerda-Kohler et al., 2022*). However, considering that the present study proposes to provide tools for the field without gas analysis, it is important to analyze some of the mechanical and HR parameters achieved in VT1. In fact, in this specific area, the pace every 500 m was $144.4 \pm 6.1$ s (62% of pace in VO$_{2max}$), while HR was equivalent to $132.9 \pm 5.5$ bpm (69% of HR$_{max}$). These results indicate that VT1 corresponds to 55 of the mechanical outcomes and 65–70% of the HR.

When analyzing VT2, it was observed that the participants reached a VO$_2$ equivalent to 80% of VO$_{2max}$. From the ventilatory perspective, the percentage value of VO$_{2max}$ reached by the rowers in this study is within the theoretical reference parameters (80–85% of VO$_{2max}$) (*López Chicharro, Vicente Campos & Cancino López, 2013*). As in VT1, it is important to observe the RER achieved in VT2 (*Davis, 1985*; *Lucía et al., 2000*; *Cerezuela-Espejo et al., 2018*). In this context, the value of RER observed in the present study for VT2 is also coincident with the RER value reported by other research in rowers (*Cerda-Kohler et al., 2022*). In this sense, it is important to observe some of the mechanical and HR parameters achieved in VT2 to use them for training prescription. In this specific zone, the pace every 500 m was $118.1 \pm 4.9$ s (85% of pace in VO$_{2max}$), while the HR was equivalent

to $162.7 \pm 10.6$ bpm (85% of $HR_{max}$). In this sense, the contribution of the present study is associated with the determination of VT. Indeed, after applying the 6-min$_{RT}$, it is possible to obtain HR and mechanical parameter values corresponding with VT2 for amateur rowers (80%). Thus, HR and mechanical parameters facilitate exercise prescription (*Cerezuela-Espejo et al., 2018*). The 6-minRT thus becomes a practical and easily accessible tool for monitoring and evaluating amateur rowers.

The $VO_{2max}$ achieved by amateur rowers classifies them as 'good' (*Herdy & Caixeta, 2016*). In addition, both the RER achieved the HR at the same intensity, indicating that the male amateur rowers reached the $VO_{2max}$ (*Davis, 1985*; *Lucía et al., 2000*; *Cerezuela-Espejo et al., 2018*). As in the VTs, the present study proposes providing tools for the field without gas analysis. Therefore, it is important to analyze some of the mechanical and HR parameters achieved in $VO_{2max}$. At this intensity, the pace every 500 m was $100.3 \pm 3.4$ s, while the HR was equivalent to $191.2 \pm 6.8$ bpm.

### 6-min$_{RT}$ and ventilatory thresholds

Previous studies have established that the 6-min$_{RT}$ is a valid and reliable test for determining MAP in male amateur rowers (*Huerta Ojeda et al., 2022b*). Likewise, the present study determined that male amateur rowers reach VT1 and VT2 at 55% and 80% of $VO_{2max}$, respectively. This background, together with the high correlations between IT outcomes and the 6-min$_{RT}$ ($r > 0.6$), allows inferring that the distance, PO, and HR achieved in the 6-min$_{RT}$ should be in good agreement with the VTs and $VO_{2max}$ of the IT. That is, 55% of the 6-minRT would correspond to VT1, 80% to VT2, and 100% to MAP. Therefore, training loads should be prescribed ≤60% (VT1) if the purpose of training is to generate intramuscular and metabolic adaptations (such as increased expression of type I myosin heavy chains, reduced work HR—sympathoadrenal adaptation—and/or optimization of the fatty acid oxidative pathway) (*López Chicharro, Vicente Campos & Cancino López, 2013*). Based on the results of the male amateur rowers in the present study, training loads should be prescribed between 55–80% (aerobic-anaerobic transition zone) to (a) increase the oxidative muscle capacity of type IIa fibers—especially the oxidation of intramuscular fats—, (b) increase the possibility of sustaining prolonged exercise with a lactate level close to 2–3 mmol-L-1, and (c) increase the buffering capacity and maintenance of musculoskeletal acid–base balance (*López Chicharro, Vicente Campos & Cancino López, 2013*). However, to achieve the physiological adaptations described above, it is suggested to train close to, but not over 80%. Finally, if the training goal is to increase muscle shortening capacity (which would lead to an increase in rowing pace), adapt the sarcoplasmic reticulum for increased calcium release, increase glycolytic activation, and enhance lactate transport across cell membranes, training loads should be prescribed 80% (*López Chicharro, Vicente Campos & Cancino López, 2013*). In this context, an Excel spreadsheet is available to automatically calculate the PO and stroke rate by zones and the pace every 500 m from the result in the 6-min$_{RT}$ (see practical applications).

Some responses and adaptations generated by the prescribed loads in the different training zones have been described. However, it is essential to mention that each athlete's physiological responses and adaptations to a stimulus are different; therefore, the training

prescription should consider the level of training, sex, and years of experience, among other factors (*Scott et al., 2016*).

### Limitations

The absence of females in the study limits the conclusions only to study participants and amateur male rowers. The lack of descriptive information on field attainment of VT1 in rowers made it challenging to compare our results.

## CONCLUSIONS

At the end of the study, it could be observed that VT1 and VT2 achieved by amateur male rowers are at 55 and 80% of $VO_{2max}$, respectively. The 6-min$_{RT}$ allows male amateur rowers to reach $VO_{2max,}$ and the outcomes of this test present a high concordance with VTs and $VO_{2max}$, evaluated through the IT (r 0.6). Consequently, to interpret and apply the 6-min$_{RT}$ results, VT1 and VT2 correspond to 55 and 80% of the mechanical and HR outcomes, respectively, while 100% of the 6-min $_{RT}$ outcomes correspond to the MAP of amateur male rowers.

### Practical applications

Considering that the 6-min$_{RT}$ allows MAP to be obtained and that, in the present study, the thresholds for this field test were determined (55% for VT1 and 80% for VT2), only the creation of tools to facilitate its use in training prescription are missing. For this purpose, a spreadsheet that automatically calculates the pace and PO times for each intensity and training zones based on the results of the 6-min $_{RT}$ is provided in this study. For the correct use of the spreadsheet, together with the personal data and evaluation date, the total distance and the average power achieved in the 6-min$_{RT}$ must be included. After this, as many sheets as necessary can be created. To facilitate the spreadsheet, 55% for VT1 and 80% for VT2 (thresholds obtained by the male amateur rowers in this study) are highlighted in yellow. The spreadsheet also allows us to project performance over 2,000 m (green cell). This projection was obtained through a linear regression performed in previous studies on male amateur rowers (*Huerta Ojeda et al., 2022a*). Finally, in black, the MAP for each rowing pace is highlighted (https://doi.org/10.6084/m9.figshare.22190026).

## ACKNOWLEDGEMENTS

We would like to thank the 16 participants who voluntarily attended the different days on which the procedures were performed.

### Funding

The authors received no funding for this work.

### Competing Interests

The authors declare there are no competing interests.

## Author Contributions

- Álvaro Huerta Ojeda conceived and designed the experiments, performed the experiments, analyzed the data, prepared figures and/or tables, authored or reviewed drafts of the article, and approved the final draft.
- Miguel Riquelme Guerra performed the experiments, authored or reviewed drafts of the article, and approved the final draft.

## Human Ethics

The following information was supplied relating to ethical approvals (i.e., approving body and any reference numbers):

The study and the informed consent were approved by the Scientific Ethical Committee of the Universidad Mayor, Santiago, Chile (registration number: 197_2020)

## Data Availability

The raw data are available in the Supplemental File and at Figshare:

Huerta, Álvaro (2023). 6-minRT spreadsheet. figshare. Dataset. https://doi.org/10.6084/m9.figshare.22190026.v1.

## Supplemental Information

Supplemental information for this article can be found online at http://dx.doi.org/10.7717/peerj.16160#supplemental-information.

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
