# Peer review of "Six-minute rowing test: a practical tool for training prescription, from ventilatory thresholds and power outputs, in amateur male rowers"

_PeerJ, doi:10.7717/peerj.16160_

## Round 0.1 · original submission · Minor Revisions

Dear authors, expert reviwers made valuable comments to improve the paper and we feel that the paper is of interest when the suggested changes are considered.
Kind regards,
Amador García-Ramos

·

Basic reporting

The reporting of the data is mostly appropriate, though with substantial repetition between the tables/figures and the text. The methods are mostly clear, though with some ommissions.
There needs to be a more detailed clarification of the use of the term 'maximal aerobic power' in the context used and in relation to the 6minute rowing test used.

Experimental design

The experimental design is appropriate and generally well presented, with some clarifications needed in some areas.

Validity of the findings

Some more detailed exploration is required in the discussion as much of it reads like a second 'mini-review of literature' and repetition of the results.

Additional comments

To Prof Ojeda and Mr Guerra,
Thank you for submitting this interesting work for review. This study has some good data and the methods appear to be appropriate, but there are some clarifications required in the methods section, as well as more specific exploration of the data in the discussion.
I look forward to reviewing the resubmission of your work.
Kind regards,
Dr Christopher Kirk

Abstract
Please check the abstract for minor spelling errors.
Introduction
Lines 54-55: Incomplete sentence, please review
Lines 63-64: Is this statement accurate, and is the term ‘maximal aerobic power’ appropriate in this context? Could the authors consider the use of the term ‘maximal aerobic power’ and whether it is appropriate in this context. The 6 minute rowing test appears to be more of a measurement of ‘maximal aerobic speed’, or as in the case of Reference 17 (Huerta et al., 2022), more to determine a rower’s maximal power whilst working at or near the maximum of their aerobic steady state. Each of these occur at a lower power than attained at V̇O2max. Whilst I’m not sure this affects the results, use of the most appropriate term for the specific variable being measured would be vital for the understanding of the results and the implications for training. This needs to be considered in the introduction, methods, and discussion.
Lines 67-72: Please include a brief statement explaining why these thresholds (VT1 and VT2) may provide important training and/or performance markers.
Lines 88-89: Please rephrase to state that ‘this study aimed to determine the power and cardio-respiratory characteristics of amateur rowers at each VT during a 6-minRT’, or something similar, as technically, VTs cannot be determined, but the measurable performance markers at which the VTs occur can be determined.

Methods
Line 106: please change ‘weight and height’ to ‘body mass and stature’ respectively in keeping with SI statements.
Lines 109-111: what was the minimum V̇O2max and hours per week of training required for inclusion? Was 55ml∙kg∙min and 6 hours∙week the minimum required or was this what the minimum participant happened to record?
Lines 112-113: Are there published guidelines for determining a rower as ‘amateur’ or ‘elite’? I suggest using the McKay et al (2021) as a framework for determining the performance standard of participants: https://journals.humankinetics.com/view/journals/ijspp/17/2/article-p317.xml
Line 134: Would it be possible to state the specific type of carbohydrate (glucose, fructose, etc) to enable accurate future replications?
Line 140: Would it be possible to give examples of what these ballistic movements were and how many repetitions were completed to enable accurate future replications?
Lines 154-155: Please double check whether the recommended RER for V̇O2max is 1.10 or 1.15 based on more recent recommendations (Cooke, 2009).
Lines 177-178: Could more specific details be provided about how the slow and fast components were visually identified? Was this done using lines of best fit to provide a quasi-V slope?
Lines 188-195: Please also state that 95% confidence intervals and R2 were calculated (as these data are reported in the results), and whether these were calculated in Excel or Prism. Please also state that correlations between variables were calculated within the IT and 6min test as well as between.

Results
Please provide the p values for all results reported in the text. The p values appear in Figures 2 and 3, but not in the text relating to these outcomes.
Figure 2: Please make it clear in the figure caption that these are correlations from the IT.
Lines 198-203: Please avoid reporting the data in the text and in the table, as this is repetition. Please just report the means±SD in the table and refer the reader to the table in the opening paragraph.
Lines 215-221: Would it be possible to combine these results into Table 1 and report the results of both the IT and the 6min test side by side for easy comparison?
Lines 230-243: Please report these data either in the text or in Table 2, as there is currently repetition here. In Table 2 it would also be helpful for the reader for all significant results to be in bold, as there are a lot of comparisons here and it’s currently difficult to pick out the relevant ones.

Discussion
Lines 248-253: Please delete these lines as they are simply repeating the results.
Lines 258-260: This is confusing – do 55% and 80% of the variables correspond to VT1 and VT2? Or do these variables explain 55% and 80% of VT1 and VT2 based on their R2 results? Please reword these sentences to make it clear what is meant here.
Lines 264-265: Why could this have occurred? What might explain this?
Lines 275-292: These paragraphs seem to be either repeating some of the results or adding new results (pacing). Please alter these sections to provide some explanation as to how/why the results occurred the way they did or why these results might be important/relevant to athletes, coaches and researchers.
Lines 294-316: My apologies, but I’m struggling to grasp the aim of this paragraph. This seems to be presenting a mini-review of the literature around non-invasive estimations of ventilatory thresholds, but it’s not clear how this relates specifically to the reported results. The final line is also confusing, as it claims that the methods used can predict VT2, but not VT1 – is this correct? As the authors report several significant correlations between measured variables at VT1 in the same way as there are several correlations between variables at VT2. Could the authors add to this section explaining how their results advance our understanding and use of the concepts discussed here.

Limitations
Line 354: Please change ‘women’ to ‘female’ to avoid reader confusion about whether the authors are discussing gender or sex.
Please focus on potential limitations of the reported data (sample size, VT1 and VT2 determination variations, etc.) rather than lack of comparable data.

Practical Applications
Does this spreadsheet exist to be downloaded and used by readers? If so, please provide a hyperlink to a usable spreadsheet. If not, then please state that coaches and other readers can use the data provided here to create their own spreadsheet. For readers to do this, they will also need to be provided regression equations from the data reported, so could the authors please provide these in the revised manuscript.

Reviewer 2 ·

Basic reporting

- Clear, unambiguous, professional English language used throughout.
- Intro & background to show context. Literature well referenced & relevant.
- Figures are relevant, high quality, well labelled & described.

Experimental design

- Research question well defined, relevant & meaningful. It is stated how the
research fills an identified knowledge gap.
- Rigorous investigation performed to a high technical & ethical standard.
- Methods need to be described in more detail and information to replicate.

Validity of the findings

- All underlying data have been provided; they are robust, statistically sound, &
controlled.
- Conclusions are well stated, linked to original research question & limited to supporting results

Additional comments

The authors present an interesting investigation in which study the ventilatory thresholds of novice rowers and their relationship with the 6-minRT. The latter parameters are essential for training prescription. I would like to congratulate the authors for conducting this interesting study. The topic is of high relevance, but the manuscript deserves to be improved. There are limitations that need to be solved. I made some minor comments to the authors with the only aim of conveying what was previously detailed.

Line 25. Please provide definition about HR. Use abbreviations every time after introducing them.

Line 26 and 104 (participants). The study entitled "Six-minute rowing test: a valid and reliable method for assessing power output in amateur male rowers" and published in PeerJ (https://doi.org/10.7717/peerj.14060) aimed to determine the validity and reliability of a 6-min rowing ergometer test had twelve participants, why does this study have sixteen participants? Why amateur rowers? Can the results be extrapolated to other levels? Indicate it in limitations section.
Little medical information about the participants is reported. Please provide information about it.

Line 46. “Another factor to consider for the good performance of the rowers is the environment” Please add reference or rewrite the sentence.

Line 97. “This time between races ensured the physical recovery of the rowers”. Please provide more information or add reference.

Line 113. Please provide information about: “those who could not perform the IT and the 6-minRT correctly”. What is correctly?

Line 126. Please provide information about the instruments (weight, height…)

Line 132. Please provide information about the blood glucose level. It was measured?

Line 140. Please provide information about the ballistic movements. Were they supervised?

Line 142. Why 10-minute rest? Justify it.

Test. How many experienced researchers controlled the test? Provide more information.

Line 187. Statistical Analysis. I would recommend reporting about the between-subject coefficient of variation to determine the variability.

Results, Discussion, Practical Applications, and Conclusions
These sections are perfectly structured and written.

---

## Round 0.2 · Minor Revisions

Authors addressed most of the comments provided by both reviewers. Small details still need to be considered in a revised version of the manuscript.

·

Basic reporting

Reporting is clear, with good English throughout and good context provided throughout.

Experimental design

Design is clear, appropriate and reproducable.

Validity of the findings

Findings are valid, with some minor amendments to make in the final interpretations for practical use.

Additional comments

To the authors,
thank you for submitting the revised manuscript for futher review. The work is much improved, with the methods now showing greater potential for replication and practical use. There are some very minor amendments to make, mainly in the final section where the data are used to make training load threshold recommendations.
I look forward to seeing this work in print and in use following these minor changes.
Kind regards,
Dr Chris Kirk

Introduction
Lines 99-100: Please clarify whether this was attained whilst completing the 6minRT, or if this was a different test that allowed the development of the 6minRT.

Discussion
Lines 387-379, lines 383-384, line 417, and line 422: VT1 was found to be 55%, not 60%. Please correct, or provide justification for using the 55-60% range.
Lines 387-388: Please justify why the aerobic-anaerobic transition zone could start at 65% based on the presented data. Why not 60%, 70%, or 75%? Suggest rewording to state that if these adaptations are sought then athletes should train close to but not above 80%.
Lines 392-393: Please change ‘running speed’ to ‘rowing pace’ or similar.
Line 395: Why >85% if VT2 = 80%? Please justify the extra 5%.
Lines 395-398: Suggest including a brief statement that the 500m paces provided by the spreadsheet will correspond to the previously discussed training threshold to make it clear for the reader.

Reviewer 2 ·

Basic reporting

No comment

Experimental design

No comment

Validity of the findings

No comment

Additional comments

I would like to thank the author for addressing most of my comments and suggestions. The only concern I still have are:

- Try to specify what is correctly “In contrast, those who could not complete the IT and the 6-minRT correctly were eliminated during the study” (line 261). Provide information about a correct execution.
- How many experienced researchers controlled the test. Please include it in the manuscript

Otherwise, I have no further remarks.

---

## Round 0.3 · accepted · Accept

The manuscript is now ready for publication. Congrats!

·

Basic reporting

Reporting is good and clear

Experimental design

The experimental design is fully described and replicable

Validity of the findings

The findings and conclusions are valid and provide a useful tool for practitioners.

Additional comments

I'm now happy that this paper now meets the required standards for publication. The results are well presented, and the resulting tool is a useful one for practitioners and sub-elite athletes to use in their training. Seeing data aimed at supporting non-elite athletes is good and provides a welcome addition to the literature. I look forward to seeing this work in print.